# Federated Heterogeneous Graph Neural Network for Privacy-preserving Recommendation

## ABSTRACT

Heterogeneous information network (HIN), which contains rich semantics depicted by meta-paths, has become a powerful tool to alleviate data sparsity in recommender systems. Existing HIN-based recommendations hold the data centralized storage assumption and conduct centralized model training. However, the real-world data is often stored in a distributed manner for privacy concerns, resulting in the failure of centralized HIN-based recommendations. In this paper, we suggest the HIN is partitioned into private HINs stored in the client side and shared HINs in the server. Following this setting, we propose a federated heterogeneous graph neural network (FedHGNN) based framework, which can collaboratively train a recommendation model on distributed HINs without leaking user privacy. Specifically, we first formalize the privacy definition in the light of differential privacy for HIN-based federated recommendation, which aims to protect user-item interactions of private HIN as well as user's high-order patterns from shared HINs. To recover the broken meta-path based semantics caused by distributed data storage and satisfy the proposed privacy, we elaborately design a semantic-preserving user interactions publishing method, which locally perturbs user's high-order patterns as well as related user-item interactions for publishing. After that, we propose a HGNN model for recommendation, which conducts node- and semantic-level aggregations to capture recovered semantics. Extensive experiments on three datasets demonstrate our model outperforms existing methods by a large margin (up to 34% in HR@10 and 42% in NDCG@10) under an acceptable privacy budget.

## CCS CONCEPTS

• **Do Not Use This Code → Generate the Correct Terms for Your Paper**.

## KEYWORDS

federated recommendation, heterogeneous information network, privacy-preserving

**ACM Reference Format:**
Anonymous Author(s). 2018. Federated Heterogeneous Graph Neural Network for Privacy-preserving Recommendation. In *Proceedings of Make sure to enter the correct conference title from your rights confirmation emai (Conference acronym 'XX)*. ACM, New York, NY, USA, 10 pages. https://doi.org/XXXXXXX.XXXXXXX

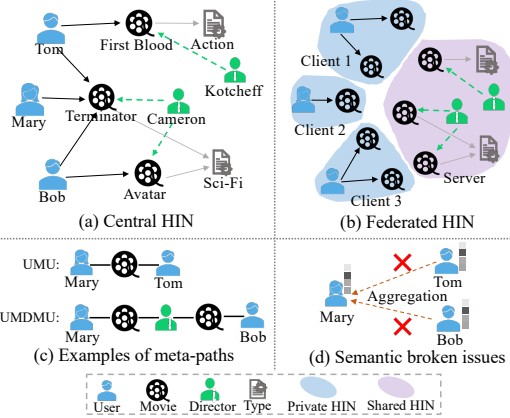

**Figure 1: Comparison of a HIN in the centralized setting and federated setting**

## 1 INTRODUCTION

Recommender systems play a crucial role in mitigating the challenges posed by information overload in various online applications [44]. However, their effectiveness is limited by the sparsity of user interactions [17, 19, 42]. To tackle this issue, heterogeneous information networks (HIN), containing multi-typed entities and relations, have been extensively utilized to enhance the connections of users and items [12, 25, 26, 41]. As a core analysis tool in HIN, *meta-path* [27], a relation sequence connecting node pairs, is widely used to capture rich semantics of HIN. Different meta-path can depict different semantics, as is shown in Figure 1, the meta-path *UMU* in the HIN for movie recommendation presents the semantics that two users have watched the same movie, and the *UMDMU* depicts that two users have watched movies directed by the same director. Most of HIN-based recommender methods leverage meta-path based semantics to learn effective user and item embeddings[11, 26]. Among them, early works integrate meta-path based semantics into user-item interaction modeling to enhance the representations of users and items [26, 41]. In recent years, graph neural networks (GNNs) have emerged as a powerful tool to capture meta-path based semantics and achieved remarkable results [6, 32, 43]. They aggregate node embeddings along meta-paths to fuse different semantics, known as meth-path based neighbor aggregation [5, 13, 31, 42], providing a more flexible framework for HIN-based recommendations.

Existing HIN-based recommendations hold a basic assumption that the data is centralized stored. As shown in Figure 1(a) and (c), under this assumption, the entire HIN is visible and can be directly utilized to capture meta-path based semantics for recommendation. However, this assumption is not always hold. The user-item interaction data is highly privacy-sensitive, and the centralized storage can leak the user privacy. Furthermore, according to the strict privacy

protection by General Data Protection Regulation (GDPR)[1], it is prohibited that commercial companies collect and exchange user data without the user's permission. In this regard, centralized data storage may be not feasible in the future.

As a more realistic learning paradigm, federated learning (FL) [21, 36] has emerged to allow all users to train a global model collaboratively without privacy leakage. Unlike the traditional centralized learning where data is stored in a central server, in FL the data is kept by its owner and not visible to others. Federated recommendation (FedRec) is an essential application of FL in the recommender scenario, in which user's original interaction data is kept locally and all users together train a global recommendation model by only transmit intermediate parameters. Many works have been dedicated to FedRec in recent years[1, 2, 15, 38]. Most of them focus on traditional matrix factorization (MF) based FedRec [2, 15]. They keep the user factors locally update and upload the gradients of item factors to the server for aggregation. Recently, a few studies have explored GNN-based FedRec [18, 20, 33]. They train local GNN models on the user-item bipartite graph and upload gradients of embedding and model parameters to the server. Despite their success, they still suffer from data sparsity issue, which might result in making inaccurate recommendations.

A natural solution is utilizing HINs to enrich the sparse interactions. However, developing HIN-based FedRec is non-trivial. It faces two challenges. 1) There lacks of a formal privacy definition of HIN-based FedRec. Compared to traditional FedRec only utilizing private information (i.e., user-item interaction), HIN-based FedRec can also utilize some shared knowledge that can be shared among users (e.g., movie-type and movie-director relations in Figure 1(a)). This shared knowledge may also leak user's high-order patterns (e.g., the user's favorite types of movies). In this regard, the privacy in HIN-based FedRec should be first clarified. 2) The meta-path based semantics are broken in HIN-based FedRec. As shown in Figure 1(b), the HIN is distributed stored and users can only access their one-hop neighbors. As a result, the integral meta-path is broken, leading to fails to conduct meta-path based neighbor aggregations, which is the key component for HIN-based recommendation. As shown in Figure 1(c) and (d), the meta-path based neighbor aggregations fail because of the broken semantics UMU and UMDMU. Therefore, it is the pain point of HIN-based FedRec to design a mechanism that recover the broken semantics and achieve privacy-preserving recommendations.

To tackle these challenges, in this paper, we study the HIN-based FedRec and propose a Federated Heterogeneous Graph Neural Network (FedHGNN) for privacy-preserving recommendations. 1) To clarify the privacy that should be protected, we propose a formal privacy definition for HIN-based FedRec. We suggest a setting for HIN-based FedRec, in which the whole HIN is divided into private HINs stored in the client side and shared HINs stored in the server. Under this setting, we formalize two kinds of privacy of HIN-based FedRec in the light of differential privacy (DP), including the privacy reflecting user's high-order patterns from shared HINs and privacy of user-item interactions with specific patterns. 2) To recover the broken semantics, we propose a semantic-preserving user interaction publishing method. Since the user's original interactions are

privacy and should not be published, we design a two-stage perturbation mechanism to perturb user interactions. Specifically, the first stage perturbs user's high-order patterns from shared HINs by exponential mechanism (EM) [4]. To preserve the user's true high-order pattern, we select patterns similar to truly user high-order patterns with high probabilities. The second stage perturbs truly user-item interactions within each selected pattern by a degree-preserving random response method[9], which avoids introducing more noise and also enhances the interaction diversity. Each user perturbs local interactions by the two-stage perturbation mechanism and uploads them to the server for recovering meta-path based neighbors. We also give rigorous privacy guarantees of the publishing process. Based on the recovered semantics, we further propose a general heterogeneous GNN model for recommendation, which captures semantics through a two-level meta-path-guided aggregation.

The major contributions of this paper are summarized as follows:

- To the best of our knowledge, this is the first work to study the HIN-based FedRec, which is an important and practical task in real-world scenarios.
- We design a FedHGNN framework for HIN-based FedRec. We give a formal privacy definition and propose a novel semantic-preserving perturbation method to publish user interactions for recommendation. We also give rigorous privacy guarantees of the publish process.
- We conduct extensive experiments on three real-world datasets, which demonstrates that FedHGNN improves existing methods by a large margin (up to 34% in HR@10 and 42% in NDCG@10) under an acceptable privacy budget.

## 2 RELATED WORK

**Heterogeneous information network based recommendation**. HIN contains rich semantics for recommendation, which has been extensively studied in recent years [10, 34, 42]. Specifically, HERec [26] utilizes a meta-path based random walk to generate node sequences and designs multiple fusion functions to enhance the recommendation performance. MCRec [11] designs a co-attention mechanism to explicitly learn the interactions between users, items, and meta-path based context. In recent years, HGNNs have been introduced for HIN modeling. To tackle the heterogeneous information, one line aggregates neighbors after transforming heterogeneous attributes into the same embedding space [12, 25]. Typically, RGCN [25] aggregates neighbors for each relation type individually. HetGNN [39] adopts different RNNs to aggregate nodes with different types. HGT [12] introduces transformer architecture [28] for modeling heterogeneous node and edge types. Another line is performing meta-path based neighbor aggregation [5, 13, 31]. HAN [31] proposes a dual attention mechanism to learn the importance of different meta-paths. HGSRec [13] further designs tripartite heterogeneous GNNs to perform shared recommendations. Unlike HAN performing homogeneous neighbor aggregation, Meirec [5] proposes a meta-path guided heterogeneous neighbor aggregation method for intent recommendation. Despite the great effectiveness of these HIN-based recommendations, they are all designed under centralized data storage and not geared for the federated setting, especially with privacy-preserving requirements.

---

[1]https://gdpr-info.eu

**Federated recommendation**. Federated learning (FL) is proposed to collaboratively train a global model based on the distributed data[7, 21, 35]. Accordingly, the global model in federated recommendation is collectively trained based on the user's local interaction data [14, 16, 22]. Each client maintains a local recommendation model and uploads intermediate data to the server for aggregation. In this process, the user's interaction behaviors (the set of interacted items or rating scores) should be protected [3]. FCF [1] is the first federated recommendation framework, which is based on the traditional collaborative filter (FCF). The user embeddings are stored and updated locally while the gradients of item embeddings are uploaded to the server for aggregation. FedMF [2] proves that the uploaded gradients of two continuous steps can also leak user privacy and thus applies homomorphic encryption to encrypt gradients. SharedMF [38] utilizes secret sharing instead of homomorphic encryption for better efficiency and FR-FMSS [15] further randomly samples fake ratings for better privacy. Recently, federated recommendations based on GNNs have emerged [18, 20, 33]. FedGNN [33] applies local differential privacy (LDP) to uploads gradient and samples pseudo-interacted items for anonymity. Besides, a trusted third-party server is utilized to obtain high-order neighbors. To perform federated social recommendations, FedSoG [18] employs a relation attention mechanism to learn local node embeddings and proposes a pseudo-labeling method to protect local private interactions. Considering personalization and communication costs, PerFedRec [20] clusters users and learned a personalized model by combining different levels of parameters. Neither these federated recommendation methods utilize the rich semantics of HINs nor have rigorous privacy guarantees.

## 3 PRELIMINARY

In this paper, we conduct HIN-based recommendation for implicit feedback. Let $U$ and $I$ denote the user set and item set. We give the related concepts as follows.

### 3.1 Heterogeneous Information Network

DEFINITION 3.1. **Heterogeneous Information Network (HIN)** [29]. A HIN $G = (V, E)$ consists of an object set $V$ and a link set $E$. It is also associated with an object type mapping function $\phi : V \to \mathcal{A}$ and a link type mapping function $\psi : E \to \mathcal{R}$. $\mathcal{A}$ and $\mathcal{R}$ are the predefined sets of object and link types, where $|\mathcal{A}| + |\mathcal{R}| > 2$.

DEFINITION 3.2. **Meta-path**. Given a HIN $G$ with object types $\mathcal{A}$ and link types $\mathcal{R}$, a meta-path $\rho$ can be denoted as a path in the form of $A_1 \xrightarrow{R_1} A_2 \xrightarrow{R_2} \cdots \xrightarrow{R_l} A_{l+1}$, where $A_i \in \mathcal{A}$ and $R_i \in \mathcal{R}$. Meta-path describes a composite relation $R = R_1 \circ R_2 \circ ... \circ R_l$ between object $A_1$ and $A_{l+1}$, where $\circ$ denotes the composition operator on relations. Then given a node $v$ and a meta-path $\rho$, the **meta-path based neighbors** $\mathcal{N}_v^\rho$ of $v$ are the nodes connecting with $v$ via the meta-path $\rho$. In a HIN, the rich semantics between two objects can be captured by the meta-path.

### 3.2 Privacy Definition

DEFINITION 3.3. **Private HIN**. A private HIN $G_p = (V_p, E_p)$ is defined as a subgraph of $G$. It is associated with an object type mapping function $\phi_p : V_p \to \mathcal{A}$ and a link type mapping function $\psi_p : E_p \to \mathcal{R}_p$, where $\mathcal{R}_p \subset \mathcal{R}$ is the set of private link types. A **user-level private HIN** contains a user $u \in V_p$ and its interacted item set $I^u \subset I$. The link set $E_p^u$ exists between $u$ and $I^u$ denoting personally private interactions.

DEFINITION 3.4. **Shared HIN**. A shared HIN $G_s = (V_s, E_s)$ is defined as a subgraph of $G$. It is associated with an object type mapping function $\phi_s : V_s \to \mathcal{A}$ and a link type mapping function $\psi_s : E_s \to \mathcal{R}_s$, where $\mathcal{R}_s$ is the set of shared link types.

As depicted in Figure 1(a) and (b), under federated setting, the movie network is divided into user-level private HINs stored in each client and shared HINs stored in the server. A user-level private HIN includes a user's private interactions and shared HINs contain shared knowledge such as movie-director relations.

A user $u$ could associate with many shared HINs based on interacted items. For example, Figure 1 (a) and (b) depict that two shared HINs are related to Tom and one shared HINs are related to Mary. These user-related shared HINs reflect high-order patterns of users (e.g., favorite types of movies) and should be protected. We call this privacy as *semantic privacy*, denoted as a user-related shared HIN list $g = (g_1, \cdots, g_{|\mathcal{G}_s|}) \in \{0, 1\}^{|\mathcal{G}_s|}$, where $\mathcal{G}_s$ denotes the whole shared HIN set. Then we formalize semantic privacy as follows:

DEFINITION 3.5. $\epsilon$-**Semantic Privacy**. Given a user-related shared HIN list $g$, a perturbation mechanism $\mathcal{M}$ satisfies $\epsilon$-semantic privacy if and only if for any $\hat{g}$, such that $g$ and $\hat{g}$ only differ in one bit, and any $\tilde{g} \in range(\mathcal{M})$, we have $\frac{Pr[\mathcal{M}(g)=\tilde{g}]}{Pr[\mathcal{M}(\hat{g})=\tilde{g}]} \leq e^\epsilon$.

Besides, each user also owns a adjacency list $a = (a_1, \cdots, a_{|I|}) \in \{0, 1\}^{|I|}$. Given $g$, we can extract a subset $I_s$ from the whole item set $I$, which is called *semantic guided item set*. Similarly, we can obtain *semantic guided adjacency list* denoted as $a_s = (a_{s1}, \cdots, a_{s|I_s|}) \in \{0, 1\}^{|I_s|}$, which depicts the user-item interactions with specific patterns and should also be protected. we call this privacy as *semantic guided interaction privacy* and formalize as:

DEFINITION 3.6. $\epsilon$-**Semantic Guided Interaction Privacy**. Given a semantic guided adjacency list $a_s$, a perturbation mechanism $\mathcal{M}$ satisfies $\epsilon$-semantic guided interaction privacy if and only if for any $\hat{a}_s$, such that $a_s$ and $\hat{a}_s$ only differ in one bit, and any $\tilde{a}_s \in range(\mathcal{M})$, we have $\frac{Pr[\mathcal{M}(a_s)=\tilde{a}_s]}{Pr[\mathcal{M}(\hat{a}_s)=\tilde{a}_s]} \leq e^\epsilon$.

$\epsilon$ is called the privacy budget that controls the strength of privacy protection. It is obvious that if a perturbation algorithm satisfies these definitions, the attacker is difficult to distinguish the user's high-order pattern as well as the true interacted items.

### 3.3 Task Formulation

Based on above preliminaries, we define our task as follows:

DEFINITION 3.7. **Federated HIN-based recommendation**. Given user-level private HINs $\mathcal{G}_p = \{G_p^{u_1}, G_p^{u_2}, ..., G_p^{u_{|U|}}\}$ and shared HINs $\mathcal{G}_s = \{G_s^1, G_s^2, ..., G_s^m\}$. The $G_p^{u_i}$ corresponding to the user $u_i \in U$ is stored in the $i$-th client, while $\mathcal{G}_s$ is stored in the server. We aim to collaboratively train a global model based on these distributed HINs with satisfying $\epsilon$-semantic privacy and $\epsilon$-semantic guided interaction privacy, which can recommend a ranked list of interested items for each user $u \in U$.

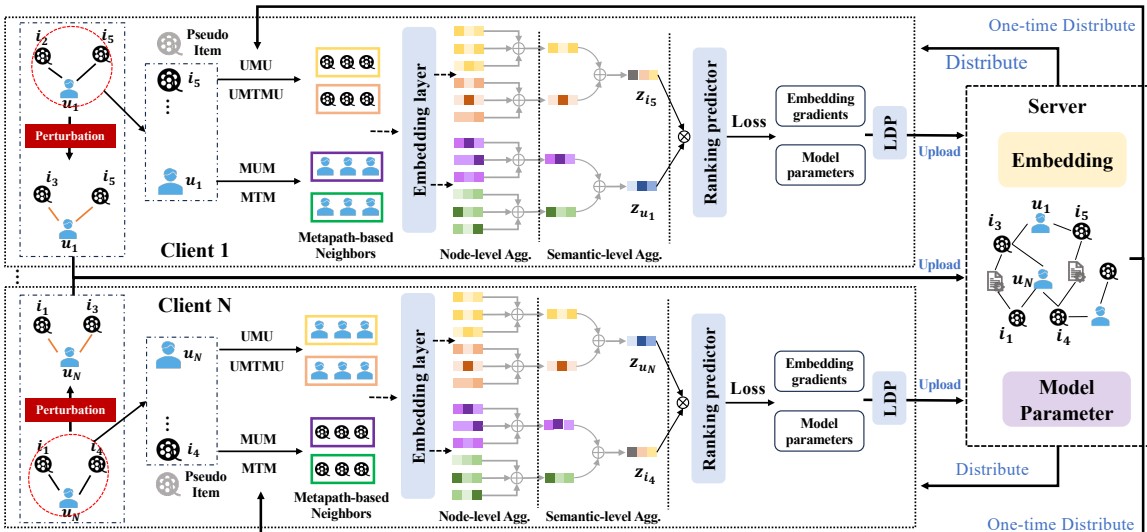

**Figure 2: The overall framework of FedHGNN**

## 4 METHODOLOGY

In this section, we give a detailed introduction to the proposed model FedHGNN. We first give a overview of FedHGNN. Then we present two main modules of FedHGNN, the semantic-preserving user-item interaction publishing and heterogeneous graph neural networks (HGNN) for recommendation. Finally we give a privacy analysis of proposed publishing process.

### 4.1 Overview of FedHGNN

Different from existing FedRec systems only utilizing user-item interactions, FedHGNN also incorporates HINs into user and item modeling, which can largely alleviate the cold-start issue caused by data sparsity. Besides, as a core component of FedHGNN, semantic-preserving user-item publishing mechanism recovers semantics with rigorous privacy guarantees, which can be applied to all meta-path based FedRec systems technically. We present the overall framework of FedHGNN in Figure 2. As can be seen, it mainly includes two steps, i.e., user-item interaction publishing and HGNN based federated training. At the user-item interaction publishing step, each client perturbs local interactions using our two-stage perturbation mechanism, and then uploads the perturbed results to the server. After the server receiving local interactions from all clients, it can form a integral perturbed HIN, which is then distributed to each client to recover the meta-path based semantics. Note that the publishing step only conduct once in the whole federated training process. At the federated training step, clients collaboratively train a global recommendation model based on recovered neighbors, which performs node-level neighbor aggregations followed by semantic-level aggregations. Then a ranking loss is adapted to optimize embedding and model parameters. At each communication round, each participated client locally trains the model and uploads the embedding and model gradients to the server for aggregations. To further protect the privacy when uploading gradients, we apply local differential privacy (LDP) to the uploaded gradients. Besides,

following previous work [18, 33], we also utilize pseudo interacted items during local training.

### 4.2 Semantic-preserving User-item Interactions Publishing

To recover the semantics of the centralized HIN (obtaining the meta-path based neighbors), directly uploading the adjacency list $a^u$ to the server can not satisfy the privacy definition because the user-item interactions are exposed. To address this, we first present a naive solution based on random response (RR) [4] and illustrate its defects of direct applications to our task. Then we give detailed introductions of our proposed two-stage perturbation mechanism for user-item interaction publishing. As depicted in Figure 3, it first perturbs the user-related shared HINs then perturbs the user-item interactions within selected shared HINs, which not only achieves semantic-preserving but also satisfies the defined privacy.

**Random response (RR)**. As many homogeneous graph metrics publishing [9, 23, 37], a straw-man approach is directly utilizing RR [4] to perturb each user's adjacency list $a^u$, i.e., the user flips each bit of $a^u$ with probability $p = \frac{1}{1+e^\epsilon}$. However, this naive strategy faces both privacy and utility limitations. For privacy, although it satisfies the $\epsilon$-semantic guided interaction privacy, it can not achieve our $\epsilon$-semantic privacy goal. As for utility, it has been theoretically proved that RR would make a graph denser [23]. Unfortunately, there exists perturbation enlargement phenomenon [40] in the HGNNs, i.e., introducing more edges may harm the HGNN's performance, which is also confirmed in our latter experiments. Besides, RR fails to accommodate the semantic-preserving since it perturbs all bits of $a^u$. We can only perturb the semantic guided item set to preserve semantics but exposing the user high-order patterns. Furthermore, the denser graph largely hinders the training speed and compounds the communication overhead in the federated setting.

**User-related shared HIN perturbation**. From the above analysis, we propose a two-stage perturbation mechanism. The first

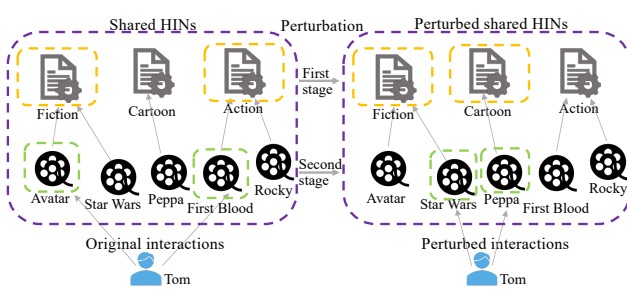

**Figure 3: The two-stage perturbation mechanism for user-item interaction publishing**

stage performs user-related shared HIN perturbation, which utilizes EM to select shared HINs for publishing. Intuitively, the true user-related shared HIN should be selected with a high probability. Therefore, according to the theory of EM, for a user $u$ with related shared HIN set $g$, we design the utility of selecting a shared HIN as follows:

$$
\begin{aligned}
q(g, u, G_s) &= sim(G_s, \mathcal{G}_s^u) \\
&= \max_{G_s' \in \mathcal{G}_s^u} \{\frac{1}{2}(cos(e_{G_s}, e_{G_s'}) + 1)\},
\end{aligned} \tag{1}
$$

where $\mathcal{G}_s^u$ is the shared HIN set of $u$ and $G_s \in \mathcal{G}_s^u$ is the selected shared HIN. $e_{G_s}$ is the representation of $G_s$ which is the average of related items' embeddings. Eq. (1) indicates that if a shared HIN $G_s$ is more similar with user-related shared HIN set $\mathcal{G}_s^u$, it should be selected with a high probability. In this regard, similarity function has multiple choices. We choose the highest cosine similarity score among $\mathcal{G}_s^u$ as the similarity function mainly in consideration of achieving a smaller sensitivity to obtain higher utility. In this way, the sensitivity $\Delta q$ is:

$$
\Delta q = \max_{G_s} \max_{g \sim \hat{g}} |q(g, u, G_s) - q(\hat{g}, u, G_s)| = 1, \tag{2}
$$

where $g \sim \hat{g}$ denotes that $g$ and $\hat{g}$ only differ in one bit. Then according to the EM, a shared HIN $G_s$ is selected with probability:

$$
\Pr(G_s) = \frac{\exp(\epsilon q(g, u, G_s)/(2\Delta q))}{\sum_{G_s' \subset \mathcal{G}_s} \exp(\epsilon q(g, u, G_s')/(2\Delta q))}. \tag{3}
$$

The above selection process is repeated $|\mathcal{G}_s^u|$ times without replacement to ensure diversity. Then we can obtain the perturbed user's shared HIN list $\hat{g}^u$. By this mechanism, the user's high-order patterns are maximum preserved since we select similar shared knowledge with high probability.

**User-item interaction perturbation**. After obtaining the perturbed $\hat{g}^u$, we can extract a semantic guided item set $I_s^u$. The user-item interaction perturbation is conducted within the $I_s^u$ rather than the whole item set. Since our $\epsilon$-semantic guided interaction privacy is defined within the $I_s^u$, ignoring the items outside of $I_s^u$ has no effect on privacy guarantees. Besides, it also avoids introducing more irrelevant items and reduces the communication cost. In light of the user-related shared HIN having been perturbed in the first stage, we can directly apply RR to perturb $I_s^u$. However, in HIN-based recommendations, the size of $I_s^u$ is still large due to

the relative small number of shared HINs, thus introducing more irrelevant items.

Inspired by [9], we propose a user-item interaction perturbation mechanism, which performs degree-preserving RR (DPRR) [9] on each of semantic guided item set. Specifically, given user $u$ and related shared HIN set $\mathcal{G}_s^u$, we can split semantic guided item set $I_s^u$ into $|\mathcal{G}_s^u|$ subsets. For each subset $I_{s_i}^u$, we use a adjacency list $a_{s_i}^u = (a_{s_i1}^u, \ldots, a_{s_i|I_{s_i}^u|}^u) \in \{0,1\}^{|I_{s_i}^u|}$ to denote the user-item interactions. DPRR perturbs each bit of $a_{s_i}^u$ by first applying RR then with probability $q_{s_i}^u$ to keep a result of 1 (a user-item interaction) unchanged. Thus the probability of each bit being perturbed to 1 is:

$$
\Pr(\tilde{a}_{s_ij}^u = 1) = \begin{cases} (1-p)q_{s_i}^u & (\text{if } a_{s_ij}^u = 1) \\ pq_{s_i}^u & (\text{if } a_{s_ij}^u = 0). \end{cases} \tag{4}
$$

Assuming the true degree of user $u$ within the subset $I_{s_i}^u$ is $d_{s_i}^u$ (i.e., the number of 1 in $a_{s_i}^u$), according to the degree preservation property [9], the $q_{s_i}^u$ should be set as follows:

$$
q_{s_i}^u = \frac{d_{s_i}^u}{d_{s_i}^u(1-2p) + |I_{s_i}^u|p}. \tag{5}
$$

In practice, the $q_{s_i}^u$ will be further clipped to $[0,1]$ to form probability. Note that the subset $I_{s_i}^u$ may not contain user-item interactions due to the perturbation on the shared HINs, in which case $q_{s_i}^u = 0$. That is, we abandon a part of the interacted items, leading to semantic losses. Instead of that, we randomly select some items within $I_{s_i}^u$ so that the total degree is equal to the true degree $d^u$. We argue that in this way the semantics of user-item interactions are preserved in light of our shared HIN selection mechanism.

## 4.3 Heterogeneous Graph Neural Networks for Recommendation

Given a recovered meta-path, our HGNN first utilizes node-level attention to learn the weights of different neighbors under the meta-path. Then the weighted aggregated embeddings are fed into a semantic-level attention to aggregate embeddings under different meta-paths. Following this process, we give an illustration of obtaining user embeddings, and item embeddings are the same.

**Node-level aggregation**. Let $h_{u_i}$ denotes the raw feature of a user $u_i$. Giving a meta-path $\rho_k$ and the recovered meta-path based neighbors $\mathcal{N}_{u_i}^{\rho_k}$ of $u_i$, the HGNN learns the weights of different neighbors via self-attention [28] followed by a softmax normalization layer:

$$
\alpha_{u_iu_j}^{\rho_k} = \text{softmax}_{u_j \in \mathcal{N}_{u_i}^{\rho_k}} (\sigma(\mathbf{a}_{\rho_k}^T \cdot [\mathbf{W}_{\rho_k} \cdot h_{u_i} || \mathbf{W}_{\rho_k} \cdot h_{u_j}])), \tag{6}
$$

where $\mathbf{W}_{\rho_k}$ and $\mathbf{a}_{\rho_k}$ are the meta-path-specific learnable parameters. Note that $\mathcal{N}_{u_i}^{\rho_k}$ only keeps the user neighbors along with the meta-path. After obtaining the attention weights, the model performs node-level aggregations to get the meta-path based user embeddings:

$$
z_{u_i}^{\rho_k} = \sigma(\sum_{u_j \in \mathcal{N}_{u_i}^{\rho_k}} \alpha_{u_iu_j}^{\rho_k} \cdot h_{u_j}). \tag{7}
$$

Since the neighbors are all in the meta-path $\rho_k$, the semantics of $\rho_k$ are fused into the user's embeddings. Thus given the meta-path set

$\mathcal{P} = \{\rho_1, \ldots, \rho_m\}$, we can obtain $m$ meta-path based embeddings $\{z_{u_i}^{\rho_1}, \ldots, z_{u_i}^{\rho_m}\}$ of $u_i$.

**Semantic-level aggregation**. User embedding with the specific meta-path only contains a single semantic (e.g., U-M-U). After we obtain user embeddings from different meta-paths, an attention-based semantic-level aggregation is conducted to fuse different semantics. Specifically, The importance (attention weights) of specific meta-path $\rho_k$ is explained as averaging all corresponding transformed user embeddings, which is learned as follows:

$$\beta^{\rho_k} = \text{softmax}_{\rho_k \in \mathcal{P}}\left(\frac{1}{|\mathcal{U}|} \sum_{u_i \in \mathcal{U}} \mathbf{q}^{\text{T}} \cdot \tanh(\mathbf{W} \cdot z_{u_i}^{\rho_k} + \mathbf{b})\right), \quad (8)$$

where $\mathbf{W}$ and $\mathbf{q}$ are the semantic-level parameters that are shared for all meta-paths and $\mathbf{b}$ is the bias vector. Then we perform semantic-level aggregations based on learned attention weights to obtain the final user embedding $z_{u_i}$:

$$z_{u_i} = \sum_{\rho_k \in \mathcal{P}} \beta^{\rho_k} \cdot z_{u_i}^{\rho_k}. \quad (9)$$

**Ranking loss**. Through the above process, we can obtain the final individual user embedding $z_{u_i}$ and item embedding $z_{v_j}$ respectively. The ranking score is defined as the inner product of them: $\hat{y}_{u_i v_j} = z_{u_i}^{\text{T}} z_{v_j}$. Then a typical bayesian personalized ranking (BPR) loss function [24] is applied to optimize the parameters:

$$L_{u_i} = - \sum_{v_j \in I^{u_i}} \sum_{v_k \notin I^{u_i}} \ln \sigma(\hat{y}_{u_i v_j} - \hat{y}_{u_j u_k}). \quad (10)$$

### 4.4 Privacy Analysis

In this section, we give a analysis of our proposed semantic-preserving user-item interactions publishing, which satisfies both $\epsilon_1$-semantic privacy and $\epsilon_2$-semantic guided interaction privacy.

THEOREM 4.1. *The semantic-preserving user-item interactions publishing mechanism achieves $\epsilon_1$-semantic privacy.*

PROOF. Let $g^u$ and $\hat{g}^u$ denote any two user-related shared HIN lists which only differ in one bit, and any output $\tilde{g}^u$ after the first-stage perturbation (denoted as $\mathcal{M}^{skp} = \{\mathcal{M}_1^{skp}, \ldots, \mathcal{M}_n^{skp}\}$ w.r.t. $n$ selections). Assuming the total privacy budget is $\epsilon_1$ and each selection consumes $\frac{\epsilon_1}{n}$ privacy budget. Since each selection is independent, we have:

$$\frac{\Pr(\mathcal{M}^{skp}(g^u) = \tilde{g}^u)}{\Pr(\mathcal{M}^{skp}(\hat{g}^u) = \tilde{g}^u)} = \frac{\Pi_{i=1}^n \Pr(\mathcal{M}_i^{skp}(g^u, q, \mathcal{G}_s) = G_{s_i})}{\Pi_{i=1}^n \Pr(\mathcal{M}_i^{skp}(\tilde{g}^u, q, \mathcal{G}_s) = G_{s_i})}$$
$$= \Pi_{i=1}^n \frac{\Pr(\mathcal{M}_i^{skp}(g^u, q, \mathcal{G}_s) = G_{s_i})}{\Pr(\mathcal{M}_i^{skp}(\tilde{g}^u, q, \mathcal{G}_s) = G_{s_i})},$$

According to the EM, we have:

$$\frac{\Pr(\mathcal{M}_i^{skp}(g^u, q, \mathcal{G}_s) = G_{s_i})}{\Pr(\mathcal{M}_i^{skp}(\tilde{g}^u, q, \mathcal{G}_s) = G_{s_i})} \leq e^{\frac{\epsilon_1}{n}},$$

Thus

$$\frac{\Pr(\mathcal{M}^{skp}(g^u) = \tilde{g}^u)}{\Pr(\mathcal{M}^{skp}(\hat{g}^u) = \tilde{g}^u)} \leq \Pi_{i=1}^n e^{\frac{\epsilon_1}{n}} = e^{\epsilon_1}. \qquad \square$$

THEOREM 4.2. *The semantic-preserving user-item interactions publishing mechanism achieves $\epsilon_2$-semantic guided interaction privacy.*

PROOF. After the first-stage perturbation, we can obtain the semantic guided item set $I_s^u$ and semantic guided adjacency list $a_s^u$ based on $\tilde{g}^u$. Let $\hat{a_s^u}$ denotes any adjacency list that only differs one bit with $a_s^u$. Without loss of generality, we assume $a_{s1}^u \neq \hat{a}_{s1}^u$. The second-stage perturbation is equivalent to first applying RR and then flipping each bit of 1 with probability $1 - q_{s_i}^u$. Denoting the RR perturbation as $\mathcal{M}^{RR}$, we have:

$$\frac{\Pr(\mathcal{M}^{RR}(a_s^u) = \tilde{a_s^u})}{\Pr(\mathcal{M}^{RR}(\hat{a_s^u}) = \tilde{a_s^u})} = \frac{\Pr(a_{s1}^u \to \tilde{a}_{s1}^u) \ldots \Pr(a_{s|I_s^u|}^u \to a_{s|I_s^u|}^{\tilde{u}})}{\Pr(a_{s1}^{\hat{u}} \to \tilde{a}_{s1}^u) \ldots \Pr(a_{s|I_s^u|}^{\hat{u}} \to a_{s|I_s^u|}^{\tilde{u}})}$$
$$= \frac{\Pr(a_{s1}^u \to \tilde{a}_{s1}^u)}{\Pr(a_{s1}^{\hat{u}} \to \tilde{a}_{s1}^u)} \leq \frac{1-p}{p}$$
$$= e^{\epsilon_2}.$$

The subsequent flipping operation can be viewed as post-processing on the $\tilde{a_s^u}$, thus the whole perturbation also achieving $\epsilon_2$-semantic guided interaction privacy. $\qquad \square$

## 5 EXPERIMENTS

### 5.1 Experimental Setup

**Datasets**. We employ three real HIN datasets, including two citation datasets (ACM and DBLP) and one E-commerce dataset (Yelp), where the basic information is summarized in Table 1. The *user* nodes and the private link types are marked in bold.

**Table 1: Dataset statistics.**

| Dataset | # Nodes | # Private/Shared Links | Meta-paths |
|---------|---------|------------------------|------------|
| ACM | **Paper (P)**: 4025
Author (A): 17431
Conference (C): 14 | **P-A**: 9703
P-C: 4025 | P-A-P
P-C-P
A-P-A |
| DBLP | **Paper (P)**: 14328
Author (A): 4057
Conference (C): 20 | **P-A**: 15368
P-C: 14328 | P-A-P
P-C-P
A-P-A |
| Yelp | **User (U)**: 8743
Business (B): 3985
Category (C): 511 | **U-B**: 11187
B-C: 11853 | U-B-U
U-B-C-B-U
B-U-B |

**Baselines**. Following [33], we compare FedHGNN with two kinds of baselines: recommendation model based on centralized data-storage (including HERec [26], HAN [31], NGCF [30], light-GCN [8], RGCN [25], HGT [12]) and federated setting for privacy-preserving (including FedMF [2], FedGNN [33], FedSog [18], PerFedRec [20]). The details of them are shown in Appendix A.

**Implementation Details**. For all the baselines, the node features are randomly initialized and the hidden dimension is set 64. We tune other hyper-parameters to report the best performance. We keep the available heterogeneous information (e.g., meta-paths) the same for all HIN-based methods. For FedGNN and FedSog, we modify the loss function as BPR loss because they originally focus on rating prediction. In FedHGNN, the learning rate is set as 0.01,

**Table 2: Overall performance of different methods on three datasets. The best result is in bold.**

| Model | | HERec | HAN | NGCF | lightGCN | RGCN | HGT | FedMF | FedGNN | FedSog | PerFedRec | FedHGNN |
|---|---|---|---|---|---|---|---|---|---|---|---|---|
| ACM | HR@5 | 0.3874 | **0.4152** | 0.3845 | 0.3684 | 0.2929 | 0.3834 | 0.0834 | 0.2608 | 0.2905 | 0.2516 | **0.3593** |
| | HR@10 | 0.4525 | 0.4727 | 0.4379 | 0.4737 | 0.4619 | **0.5035** | 0.1331 | 0.345 | 0.3642 | 0.3229 | **0.4185** |
| | NDCG@5 | 0.3222 | **0.335** | 0.322 | 0.2624 | 0.1752 | 0.2612 | 0.056 | 0.193 | 0.2201 | 0.1824 | **0.2787** |
| | NDCG@10 | 0.3333 | **0.3537** | 0.3393 | 0.2968 | 0.2302 | 0.3001 | 0.072 | 0.2202 | 0.2438 | 0.2055 | **0.298** |
| DBLP | HR@5 | 0.3265 | **0.3877** | 0.3161 | 0.3256 | 0.387 | 0.3252 | 0.0998 | 0.2301 | 0.1978 | 0.1676 | **0.3376** |
| | HR@10 | 0.3882 | 0.4498 | 0.3895 | 0.4419 | **0.5074** | 0.4763 | 0.1606 | 0.3252 | 0.2691 | 0.2619 | **0.4373** |
| | NDCG@5 | 0.2586 | **0.33** | 0.246 | 0.2281 | 0.2763 | 0.2264 | 0.0603 | 0.167 | 0.14 | 0.105 | **0.2481** |
| | NDCG@10 | 0.2717 | **0.3503** | 0.27 | 0.2646 | 0.3151 | 0.2748 | 0.0732 | 0.1963 | 0.163 | 0.1352 | **0.2778** |
| Yelp | HR@5 | 0.2322 | 0.2877 | 0.1831 | 0.2368 | 0.2844 | **0.3322** | 0.0712 | 0.1801 | 0.1839 | 0.1513 | **0.2178** |
| | HR@10 | 0.3322 | 0.4077 | 0.2958 | 0.3684 | 0.3907 | **0.4635** | 0.1259 | 0.2596 | 0.2715 | 0.237 | **0.2977** |
| | NDCG@5 | 0.1637 | 0.1929 | 0.1127 | 0.1881 | 0.2003 | **0.2311** | 0.0444 | 0.1221 | 0.1227 | 0.1002 | **0.1578** |
| | NDCG@10 | 0.1961 | 0.2316 | 0.1493 | 0.2307 | 0.2346 | **0.2733** | 0.0619 | 0.1477 | 0.1508 | 0.1277 | **0.1834** |

$\epsilon_1$ and $\epsilon_2$ are all set as 1. For each dataset, we first perform item clustering based on shared knowledge so that each item only belongs to one shared HIN. The number of shared HIN (number of clustering) is set as 20 for all datasets. The number of attention head is set to 2 and we set early stopping if there is no improvement for 40 epochs. For LDP and pseudo interacted items, we set the hyper-parameters as the same with [18, 33]. Following [20], we apply the leave-one-out strategy for evaluation and use HR@K and NDCG@K as metrics.

## 5.2 Overall Performance

Table 2 shows the overall results of all baselines on three datasets. The following findings entail from the Table 2: (1) FedHGNN outperforms all the federated recommendation models by a big margin (up to 34% in HR@10 and 42% in NDCG@10), which demonstrates the effectiveness of our model. Surprisingly, FedHGNN also outperforms several centralized models (notably non-HIN based methods, e.g., NGCF), which is attributed to that more heterogeneous information is utilized. (2) Among centralized baselines, HIN-based methods perform better, especially on sparse datsets (e.g., dblp), owing to introducing additional semantic information to alleviate cold-start issue. It has also been observed that HAN achieves better results than other HIN-based methods, indicating that meta-path based neighbor aggregation may be more potent than non-GNN based methods (HREec) and heterogeneous neighbor aggregation methods (RGCN and HGT). Therefore, we choose HAN as our base recommendation model. (3) Among federated baselines, FedMF performs poorly because it ignores the high-order interactions which is significant for cold-start recommendation. The other three federated models improve this by privacy-preserving graph expansion (FedSog assumes social relation is public). In contrast, our FedHGNN further considers semantic information with theoretically guaranteed privacy protection.

## 5.3 Ablation Study

To have a in-depth analysis of our two-stage perturbation mechanism, we conduct ablation studies to dissect the effectiveness of different modules. We design 7 variants based on FedHGNN and

the performance of these variants is outlined in Table 3. *FedHGNN** is the FedHGNN model without two-stage perturbation. *RR* means random response and *DPRR* is the degree-preserving RR. *+S* indicates adding corresponding perturbation to each semantic guided adjacency list $a^u_{s_i}$, otherwise to each user's adjacency list $a^u$. Note that *SDPRR** indicates performing DPRR to the whole semantic guided adjacency list $a^u_s$, which is the only difference with our FedHGNN. *+E* indicates adding EM perturbation. We set $\epsilon_2 = \epsilon_2 = 1$ for all variants except that $\epsilon_2 = 6$ for RR-related variants, due to a smaller $\epsilon_2$ makes the graph denser, which sharply increases training time and consumed memory.

From the table we have several findings: (1) After two-stage perturbation of adjacency lists, the performance of FedHGNN is even superior to the model without perturbation. We find that the number of user-item edges slightly increase after perturbation. Considering the datasets are relatively sparse, we assume the perturbation can be seen as an effective data augmentation method to alleviate cold-start recommendation issues. (2) Pure RR and DPRR perform poorly since they perturb user-item interactions randomly without considering semantic-preserving. Pure RR perform even worse due to it makes a graph denser and cause perturbation enlargements [40]. DPRR preserves degrees but fail to preserve user-item interaction features. Thus we can draw a conclusion that semantic-preserving requires both degree-preserving and feature-preserving. On the contrary, perturbation within the semantic guided item set (*+SRR* and *+SDPRR*) performs much better, which further verifies our conclusion. (3) Adding first-stage perturbation (EM) will harm the performance but is necessary, otherwise we can not protect the user high-order patterns. Thanks to our designed similarity-based EM, the performance has not decreased dramatically. Note that FedHGNN also outperforms *+ESDPRR**, indicating we should keep the diversity of user-item interactions after EM, i.e., the interacted items should exist in each selected shared HIN.

## 5.4 Parameter Analysis

In this section, we investigate the impacts of some significant parameters in FedHGNN, including the number of shared HINs, as well as the privacy budgets $\epsilon_1$ and $\epsilon_2$.

Table 3: Performance of different variants of FedHGNN on three datasets.

| | Model | FedHGNN* | +RR | +DPRR | +SRR | +SDPRR | +E | +ESRR | +ESDPRR* | FedHGNN |
|---|---|---|---|---|---|---|---|---|---|---|
| ACM | HR@5 | 0.3118 | 0.0495 | 0.1749 | 0.3437 | 0.389 | 0.3461 | 0.2959 | 0.3475 | 0.3593 |
| | HR@10 | 0.3961 | 0.1118 | 0.2268 | 0.4998 | 0.5027 | 0.4004 | 0.3861 | 0.4069 | 0.4185 |
| | NDCG@5 | 0.2293 | 0.0345 | 0.1326 | 0.2312 | 0.2865 | 0.266 | 0.215 | 0.266 | 0.2787 |
| | NDCG@10 | 0.2567 | 0.0491 | 0.1492 | 0.2845 | 0.323 | 0.2835 | 0.2438 | 0.2852 | 0.298 |
| DBLP | HR@5 | 0.2824 | 0.0694 | 0.1678 | 0.2224 | 0.3346 | 0.2616 | 0.2729 | 0.3156 | 0.3376 |
| | HR@10 | 0.3934 | 0.1237 | 0.2394 | 0.3154 | 0.4557 | 0.3701 | 0.3718 | 0.4227 | 0.4373 |
| | NDCG@5 | 0.2176 | 0.0429 | 0.1115 | 0.1458 | 0.2484 | 0.1835 | 0.1929 | 0.2273 | 0.2481 |
| | NDCG@10 | 0.241 | 0.0602 | 0.1413 | 0.1757 | 0.2801 | 0.2176 | 0.2249 | 0.2619 | 0.2778 |
| Yelp | HR@5 | 0.2583 | 0.0663 | 0.1383 | 0.1172 | 0.2364 | 0.2244 | 0.223 | 0.1871 | 0.2178 |
| | HR@10 | 0.3482 | 0.1232 | 0.2079 | 0.1803 | 0.3245 | 0.3242 | 0.3257 | 0.2624 | 0.2977 |
| | NDCG@5 | 0.1859 | 0.0392 | 0.0963 | 0.0672 | 0.171 | 0.152 | 0.1538 | 0.1321 | 0.1578 |
| | NDCG@10 | 0.2201 | 0.0575 | 0.1185 | 0.0789 | 0.1976 | 0.1875 | 0.1804 | 0.1563 | 0.1834 |

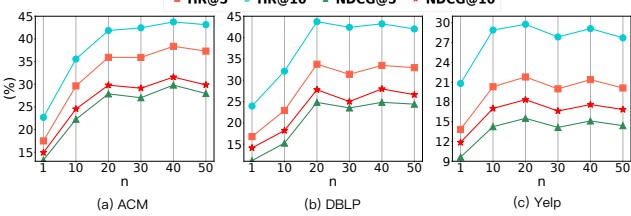

(a) ACM     (b) DBLP     (c) Yelp

Figure 4: Effects of different number $n$ of shared HINs

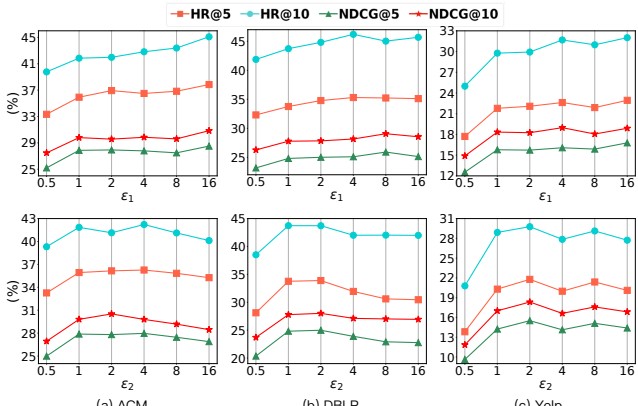

(a) ACM     (b) DBLP     (c) Yelp

Figure 5: Effects of different privacy budget $\epsilon_1$ and $\epsilon_2$

**Analysis of different number of shared HINs**. To demonstrate the effects of different number $n$ of shared HINs, we fix other hyper-parameters unchanged and vary $n$ to compare the performance. The results are depicted in Figure 4. Considering two extreme conditions: when $n = 1$, the two-stage perturbation degenerate to solely second-stage perturbation, i.e., perturbation by DPRR on the whole item set, which fails to preserve user-item interaction patterns as discussed in Section 5.3; When $n = |I|$, according to Eq. 5, it is equivalent to performing RR in each 1's bit after the first-stage perturbation, which intuitively perform better than $n \leq |I|$.

According to this theory, the performance will increase when $n$ is larger. However, as can be seen, the performance of all datasets has a dramatic incremental trend at the initial stage of increasing $n$ ($n \leq 20$), then the curve becomes smooth and even has a decrement trend ($n \geq 20$). We attribute this phenomenon to that the model with perturbed user-item interactions is performing better than with true interactions, as depicted in Table 3, and a large $n$ will reduce this effects. In summary, $n$ controls the trade-off between utility and privacy, a larger $n$ may bring relatively higher utility but weaker privacy protection, since the attacker can conclude the user-item interactions within a small scope.

**Analysis of different privacy budget**. To analyze the effects of different $\epsilon_1$ and $\epsilon_2$, we fix one parameter as 1 and change another one from 0.5 to 16 to depict the performance in Figure. 5. $\epsilon_1$ controls the protection strength of user behavior patterns (related-shared HINs). We can see that the metrics gradually increase with $\epsilon_1$, indicating the user behavior patterns are significant for recommendation and this patterns are undermined when $\epsilon_1$ is too small (e.g., 0.5). When fixing $\epsilon_1 = 1$, the performance curve of $\epsilon_2$ will first increase then slightly decrease. We suppose that due to the user behavior patterns are already perturbed in the first stage, the second stage perturbation is conducted on a contaminated interactions, thus the performance may still drop when $\epsilon_2$ is large. It also shows that conducting moderate perturbation will promote the performance (e.g., $\epsilon = 1$).

# 6 CONCLUSION

In this paper, we first explore the challenge problem of HIN-based federated recommendation. We formulate the privacy in federated HIN and propose a semantic-preserving user-item publishing method with rigorous privacy guarantees. Incorporating this publishing method into advanced heterogeneous graph neural networks, we propose a FedHGNN framework for recommendation. Experiments show that the model achieves satisfied utility under an accepted privacy budget.

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

# A  DESCRIPTION OF BASELINES

The detailed descriptions of baselines are presented as follows:

- **HERec** [26] is a HIN-enhanced recommender framework based on matrix factorization. It first trains HIN-based embeddings then incorporates them into matrix factorization.
- **HAN** [31] introduces meta-path based neighbor aggregation to learn node embeddings. We utilize the learned embeddings to perform recommendation in an end-to-end fashion.
- **NGCF** [30] is a GNN-based recommender method which learns embedding via message passing on user-item bipartite graph.
- **lightGCN** [8] improves NGCF by removing feature transformation and nonlinear activation.
- **RGCN** [25] utilizes GCN to learn node embeddings of knowledge graph. It performs message-passing along different relations.
- **HGT** [12] proposes a transformer architecture for HIN modeling, which conducts heterogeneous attention when aggregating neighbors.
- **FedMF** [2] is a federated recommender framework based on matrix factorization. The user embedding is updated locally and encrypted item gradient is uploaded to the server for aggregation.
- **FedGNN** [33] is a GNN-based federated recommender framework. It obtains high-order user neighbors through a third-party trustworthy server and utilizes pseudo-interacted item sampling to achieve privacy-preserving.
- **FedSog** [18] is another GNN-based federated recommender framework. It proposes a relational graph attention network to perform social recommendations.
- **PerFedRec** [20] is a personalized federated recommender method. It performs user clustering on the server and then aggregates parameters within each cluster to achieve personalization.

