# OpenReview forum: "Federated Heterogeneous Graph Neural Network for Privacy-preserving Recommendation"
_ACM.org/TheWebConf/2024/Conference — TheWebConf24 Oral_

### Official Review · Reviewer_dLDZ · 2023-11-09

**Novelty:** 7
**Technical Quality:** 7

**Review:**

This paper introduces FedHGNN, a privacy-preserving federated recommender system that harnesses a heterogeneous information graph to predict user preferences. FedHGNN integrates a privacy-aware interaction publishing method to protect user interaction data within two stages: an explosion mechanism-based sampling followed by a degree-preserving perturbation. Extensive experiments conducted on three datasets demonstrate the effectiveness of FedHGNN.

The strengths and weaknesses of this paper are as follows.

Strengths:

S1. The research topic is both intriguing and pivotal within federated recommender systems. Understanding how to effectively utilize high-order graph information within the realm of FedRec is of utmost importance.

S2. The proposed method is easily comprehensible, presenting a clear and logical technical description.

S3. The experimental results are convincing to support the superiority of FedHGNN.

Weaknesses:

W1. In the preliminary part, the concept of “semantic”, “semantic-guided”, “user-related shared graph” are not very clear. The authors are encouraged to introduce the technical concept based on examples to improve this part’s readability.

W2. What is the final privacy budget when combining two stages (EM +DPRR)?

W3. It would be better to use bold to highlight the best values in Table 3.

W4. Some references about privacy-preserving FedRec, graph-based FedRec, and heterogeneity can be introduced in the related work part.

[1] Interaction-level Membership Inference Attack Against Federated Recommender Systems

[2] HeteFedRec: Federated Recommender Systems with Model Heterogeneity

[3] Semi-decentralized Federated Ego Graph Learning for Recommendation

[4] Comprehensive privacy analysis on federated recommender system against attribute inference attacks

Generally, this paper is well-written, the proposed method is novel, and the experiments are comprehensive. I would like to recommend an acceptance.

**Questions:**

See the weaknesses W1-W4.

**Ethics Review Description:**

None.

**Reviewer Confidence:**

4: The reviewer is certain that the evaluation is correct and very familiar with the relevant literature

**Scope:**

4: The work is relevant to the Web and to the track, and is of broad interest to the community

---

### Official Review · Reviewer_5C1J · 2023-11-21

**Novelty:** 6
**Technical Quality:** 6

**Review:**

To address the privacy concern in graph-based federated recommender systems, this paper proposes a federated heterogeneous graph neural network (FedHGNN) framework that partitions the HIN into private HINs stored on the client side and shared HINs on the server. They define privacy in the context of HIN-based federated recommendation and propose a semantic-preserving user interactions publishing method to recover broken meta-path based semantics caused by distributed data storage. Experimental results demonstrate that the FedHGNN model outperforms existing methods while maintaining an acceptable privacy budget.

Pros:
1. The motivation of this paper, i.e., leveraging HINs to improve the performance of GNN-based FedRec , is well introduced and meaningful to me.
2. The proposed method is technically sound, especially the proposed semantic-preserving user interaction publishing method. This method effectively addresses the key technical challenge of recovering the broken semantics in the context of graph-based FedRec.
3. The experiments are comprehensive, and experimental results demonstrate the effectiveness of the proposed method.


Cons:
1. Some works on GNN-based FedRec are missing, such as [1]. The authors should discuss the research gaps between the proposed method and the missing related works.
2. Communication costs are also key metric in federated recommendation, but the corresponding results are missing.

Typos:
1. In line 296 on page 3, where it says "defined as a subgraph ofG", there is a missing space between "of" and "G".


Reference:

[1] Qu, L., Tang, N., Zheng, R., Nguyen, Q. V. H., Huang, Z., Shi, Y., & Yin, H. (2023). Semi-decentralized Federated Ego Graph Learning for Recommendation. WWW 2023.

**Questions:**

1. What is the difference between the proposed method and the missing related works?
2. How does the communication cost of the proposed method compare with other methods?

**Reviewer Confidence:**

4: The reviewer is certain that the evaluation is correct and very familiar with the relevant literature

**Scope:**

4: The work is relevant to the Web and to the track, and is of broad interest to the community

---

### Official Review · Reviewer_37sH · 2023-11-23

**Novelty:** 5
**Technical Quality:** 3

**Review:**

### Summary
This work studies the heterogeneous information network (HIN) in the setting of federated learning (FL), where the data in local clients may not be accessible to others, and, particularly in the graph FL setting, the edges between graphs of different users are disconnected. To tackle those issues, the authors formulate the privacy definitions of HIN models in the FL setting, and then propose the graph sharing scheme (semantic-preserving user-item interactions publishing), which perturbs the original graph in order to mitigate the privacy concerns in data sharing. The authors evaluate the proposed method, namely FedHGNN, on multiple datasets, showing its effectiveness.

---

### Strengths
* This work extends the existing graph FL, which mostly considers simple homogeneous graphs or heterogeneous graphs without considering their meta-paths, and handles the heterogeneous graphs with the meta-path-based approach in FL.
* The proposed method (semantic-preserving user-item interactions publishing) is interesting, for tackling the problem of disconnected edges between local subgraphs.

---

### Weaknesses
* The novelty of this work on the problem side is limited. Existing work, namely FedGNN cited in this paper, already studies 1) the privacy definition on graph FL (where edges are disconnected across local subgraphs) and 2) the method for handling this issue.
* Even after perturbing the original graphs with the proposed approach in Section 4.2, the perturbed graphs may have the information of local users (i.e., some nodes and edges remain the same), which are sent to the server, which may violate the privacy constraint of FL (local data should be only locally accessible).
* The results in Table 3 should be further clarified. First of all, for the Yelp dataset, the simple FedHGNN* model, which performs FL with the HIN-based method, outperforms the proposed full FedHGNN that consists of many proposed components. In this regard, one can think that the performance improvements of the proposed FedHGNN in the main results (Table 2) may come from using the HIN-based method in the FL setting, and not from using the proposed components. Also, FedHGNN* with SDPRR or E brings performance improvements. However, when using both of them (SDPRR and E) together, the performances largely decrease on ACM and DBLP datasets, which are unclear.
* Some previous works [A, B] tackle the problem of disconnected edges between local graphs in the graph FL setting, which are worthwhile to discuss, since this work also tackles the same problem, but with a different approach: semantic-preserving user-item interactions publishing.
* How to preserve the semantics of graphs after perturbation (Lines 445) is not clear. The perturbation operation may alter the semantics of the original graphs.

---

In summary, while the main proposed idea (perturbation of user-item interactions for sending that information to the server in order to capture each user's semantics) is interesting and somewhat novel, the concerns about its validity in FL and its experimental effectiveness exist.

---

[A] Subgraph Federated Learning with Missing Neighbor Generation, NeurIPS 2021.

[B] Personalized Subgraph Federated Learning, ICML 2023.

**Questions:**

Please see the main review above. The point below is a minor suggestion/question.
* After perturbation operations, how many nodes and edges (on average) remain the same? If only a small number of nodes and edges is changed after perturbation, there may exist a risk of privacy leakage as many nodes and edges of the users' local graphs are sent to the server. On the other hand, if they are changed largely, the perturbed graphs may significantly lose the semantics of their corresponding original graphs, which may not be useful in FL.

**Reviewer Confidence:**

3: The reviewer is confident but not certain that the evaluation is correct

**Scope:**

3: The work is somewhat relevant to the Web and to the track, and is of narrow interest to a sub-community

---

### Official Review · Reviewer_EqNz · 2023-11-23

**Novelty:** 6
**Technical Quality:** 5

**Review:**

Pros: The authors are pioneering to explore the problem of HIN-based federated recommendation. They formulate the privacy in federated HIN and propose a semantic-preserving user-item publishing method with rigorous privacy guarantees. They design a FedHGNN model for recommendations. They conduct extensive experiments and theoretical proof to verify its effectiveness.
Cons: There may exist some flaws in the experiment settings (see questions).

**Questions:**

1.	Why do the authors use randomly initialized features when implementing the baseline methods, while it seems the FedHGNN uses the raw node features? Does this make an unfair comparison between them?
2.	In the experiment setting part, the authors mentioned they perform item clustering first. What does this ‘item’ mean here? The ACM and DBLP datasets are academic networks, not recommendation datasets.
3.	The partition of the shared HIN is ambiguous.
4.	The model is designed for recommendation while it uses only one related dataset.

**Ethics Review Description:**

NA.

**Reviewer Confidence:**

4: The reviewer is certain that the evaluation is correct and very familiar with the relevant literature

**Scope:**

4: The work is relevant to the Web and to the track, and is of broad interest to the community

---

### Official Review · Reviewer_87rd · 2023-11-23

**Novelty:** 4
**Technical Quality:** 4

**Review:**

This paper introduces a federated HIN model and explores its application in the recommendation scenario. Specifically, it begins by defining the privacy considerations in federated HIN and proposes a semantic-preserving strategy as a solution. It is important to note that while the proposed method focuses on federated HIN, it does not directly address federated recommendation.

Pros:
1. The proposed method demonstrates superior performance compared to most baseline models.

Cons:
1. The related work summary and experimental verification lack existing federated recommendation models, e.g., [1][2]. This omission weakens the overall context and understanding of the research.

[1] Zhang C, Long G, Zhou T, et al. Dual Personalization on Federated Recommendation. IJCAI, 2023.

[2] Qu L, Tang N, Zheng R, et al. Semi-decentralized Federated Ego Graph Learning for Recommendation. WWW, 2023.

2. The novelty of the proposed method is limited, and the research challenge is not clearly defined. Additionally, the definition of semantic and interaction privacy can be confusing.
3. Uploading user embeddings to the server raises concerns about user privacy leakage. In general federated recommendation studies, user embeddings are considered private modules that remain locally preserved and are not exposed to others.
4. The selected datasets used in the experiments are not specifically tailored for the recommendation task, which undermines the persuasiveness of the results.

**Questions:**

1. How to define the shared HIN for each user? If it is based on the user's interacted items, the definition of the shared HIN could potentially compromise user privacy.
2. How many clients should participate in the federated optimization during each communication round? Involving all clients would result in significant communication overhead, making it impractical for real-world application scenarios.
3. Why does the proposed method outperform certain centralized baselines? The reasons behind the improved performance of the proposed method need to be clearly explained and substantiated.

**Ethics Review Description:**

N.A.

**Reviewer Confidence:**

4: The reviewer is certain that the evaluation is correct and very familiar with the relevant literature

**Scope:**

3: The work is somewhat relevant to the Web and to the track, and is of narrow interest to a sub-community

---

### Decision · Program_Chairs · 2024-01-22

**Decision:**

Accept (Oral)

**Comment:**

This paper presents an exploration of federated recommendation, introducing a novel approach known as the Federated Heterogeneous Graph Neural Network (FedHGNN) framework to enhance the performance of Graph Neural Network (GNN)-based Federated Recommendation (FedRec) through the utilization of HINs. The rationale behind this focus is well-illustrated and articulated, providing a clear understanding of the motivation driving the research. An intriguing aspect of the paper is its exploration of privacy within the context of HIN-based federated recommendations. Moreover, the proposed method for semantic-preserving user interaction publishing, aimed at recovering broken meta-path-based semantics, is both innovative and compelling. The experimental results presented in the paper are promising, indicating the potential effectiveness of the FedHGNN framework. This empirical validation enhances the credibility of the proposed approach and underscores its practical viability.


 Nevertheless, there are opportunities for enhancing the paper in several key aspects. Firstly, the absence of references to some recent works on Graph Neural Network (GNN)-based Federated Recommendation (FedRec) is notable. It is strongly recommended that the authors conduct a comprehensive analysis, both analytically and empirically, to compare their work with these missing ones. The clarity of the preliminary section could be improved, particularly in relation to concepts such as "semantic," "semantic-guided," and "user-related shared graph." To enhance readability, it is advisable to introduce these technical concepts through illustrative examples, ensuring a more accessible understanding for readers. Moreover, the choice of datasets warrants attention. While the paper primarily focuses on the recommendation task, the use of more classic recommendation datasets is encouraged instead of those conventionally employed for Heterogeneous Information Network (HIN) tasks.